# The Impact of Molecular Biology in the Seeding, Treatment Choices and Follow-Up of Colorectal Cancer Liver Metastases—A Narrative Review

**DOI:** 10.3390/ijms24021127

**Published:** 2023-01-06

**Authors:** Mihai-Calin Pavel, Elena Ramirez-Maldonado, Eva Pueyo-Périz, Robert Memba, Sandra Merino, Justin Geoghegan, Rosa Jorba

**Affiliations:** 1HPB Unit, Department of General Surgery, Hospital Universitari de Tarragona Joan XXIII, 43005 Tarragona, Spain; 2Departament of Medicine and Surgery, Universitat Rovira i Virgili, 43204 Reus, Spain; 3Oncology Department, Hospital Universitari de Tarragona Joan XXIII, 43005 Tarragona, Spain; 4Oncology Department, Hospital Sant Joan de Reus, 43204 Tarragona, Spain; 5HPB and Liver Transplant Surgery Department, St. Vincent’s University Hospital, D04 T6F4 Dublin, Ireland

**Keywords:** colorectal cancer liver metastasis, molecular biology, monoclonal antibodies, follow-up

## Abstract

There is a clear association between the molecular profile of colorectal cancer liver metastases (CRCLM) and the degree to which aggressive progression of the disease impacts patient survival. However, much of our knowledge of the molecular behaviour of colorectal cancer cells comes from experimental studies with, as yet, limited application in clinical practice. In this article, we review the current advances in the understanding of the molecular behaviour of CRCLM and present possible future therapeutic applications. This review focuses on three important steps in CRCLM development, progression and treatment: (1) the dissemination of malignant cells from primary tumours and the seeding to metastatic sites; (2) the response to modern regimens of chemotherapy; and (3) the possibility of predicting early progression and recurrence patterns by molecular analysis in liquid biopsy.

## 1. Introduction

Today, colorectal cancer (CRC) is the second most common cause of death related to cancer, usually due to distant metastasis [1]. More than 50% of patients diagnosed with CRC will present at some point in the course of their disease with metastases [2]; of these 20–30% will be confined exclusively to the liver (CRCLM) [3].

Liver resection is the most effective curative option for patients affected by CRCLM [2,4,5], with a 5-year overall survival rate of 28–58% [6,7,8,9,10]. However, more than 80% of patients with CRCLM will have unresectable disease at the time of diagnosis [2,11,12].

Chemotherapy represents an integral part of the multidisciplinary treatment of CRCLM patients. In general, chemotherapy may be useful in three different scenarios: up-front resectable metastatic disease (especially synchronous CRCLM) [13], potentially resectable metastases and non-resectable (palliative) disease [14]. Each one of these scenarios has its own objectives, depending on which the chemotherapy regimen may vary. The introduction of monoclonal antibodies in addition to standard chemotherapy has increased the resection rates of initially unresectable CRCLM but also highlights the importance of the molecular tumoural pattern in the choice of a treatment [15,16].

The development of metastases is related to the complex molecular pathways that direct the evolution of the cancer cells. These pathways are involved in three important aspects of the natural history and the prognosis under the treatment of these tumours: first, the dissemination of malignant cells from the primary tumour to metastatic sites (related not only to the tumour cells but also to the favourable hepatic environment for metastatic implantation—the so-called “metastatic niche”) [17]; second, the molecular pattern of the liver metastases that is associated with the selection of the chemotherapy to be administered [4,5]; and third, the evaluation of the response to treatment and the use of molecular markers that are predictive of recurrence, which may be used in follow-up.

To date, the impact of the molecular profile of tumours in these three stages of oncological disease is still a matter of debate. The objective of this article is to investigate the current literature related to the molecular pathways of CRCLM involved in the seeding of the metastasis, their use in selection of chemotherapy and the response to treatment, and the identification of recurrence.

## 2. Metastatic Niche

The dissemination of malignant cells from primary tumours to metastatic sites is a key step in cancer progression. Currently, the molecular mechanisms underlying the first steps of CRCLM formation remain unclear. However, recent investigations have shown the importance of the interaction between the tumoural cells and the so-called extracellular matrix (ECM) in each step of the development of liver metastasis. ECM is defined as the collection of extracellular proteins that provide the three-dimensional scaffold within which cells organize to form complex structures. For circulating tumour cells (CTCs), the hepatic ECM provides a microenvironment where they can survive and proliferate [17].

The ECM plays an important role in each step of metastasis development—survival in the blood stream, seeding in the liver and proliferation within the organ.

CTCs seed in specific target organs. The development of such metastases depends on the dynamic interactions that take place between the cancer cells and the microenvironments found in the target organ. The liver is, of course, a particular target for the metastatic spread of CRC. There are currently two emerging theories related to the biology of metastasis. First, organ-specific microenvironments could influence the fate of disseminated cancer cells that reach the liver. Second, a more dynamic concept suggests that disseminated cancer cells may interact with the ECM, altering it to create a more favourable environment for the development of metastasis. Such interactions between the healthy and premetastatic liver matrix may facilitate the early seeding of disseminated cancer cells. At the same time, the matrix derived from both cancer and liver contributes to changes in the composition of the “niche” as the disease progresses [17].

Several studies suggest that the primary tumour may influence homeostasis in distant organs with which they share circulation even before the metastasis appears (a concept known as “premetastatic niche”) [17,18]. Experimental studies have proven that the livers of tumour-affected mice have enhanced deposits of fibronectin and collagen I as well as RNA alteration of transcription for several proteins prior to the seeding of liver metastasis [18]. The signals from the primary tumour seem to hijack normal homeostatic pathways related to early inflammation as a response to injury or infection [19].

The fate of CTCs in the bloodstream seems to be related to the capacity for seeding in distant organs. Less than 0.02% of CTCs generate metastases. The rest are cleared from the circulation by immune interaction or anoikis [20]. One mechanism by which cellular death may be avoided is the association of CTCs in clusters or with blood cells, a process that is related to intense ECM production [19,21]. 

Furthermore, the capacity of CTCs to penetrate liver sinusoids is related to ECM proteins. Cancer-derived genes, such as SERPINB5 and CSTB, support extravasation and are overexpressed in metastatic cancers [22]. Once the circulating tumour cells are established as metastatic deposits inside the liver, they produce changes in the cellular and extracellular composition of the organ. A study with colorectal cancer cells identified 56 proteins upregulated in the metastatic liver that were not expressed in the normal liver or in the primary tumour [23]. Tumour cells of different origins produce different patterns of protein production. Even the growth pattern of the metastasis differs depending on the type of tumour. Colorectal cancer metastases generate a barrier of desmoplastic extracellular matrix between the tumour and the healthy liver, which helps the tumour growth and has a more disruptive effect on the normal liver [24].

There are multiple lines of research in this regard, since it is a topic of great clinical importance. There are studies reporting a novel role for homeobox B5 (HOXB5), a member of the HOX family, in promoting CRC metastasis. Elevated expression of the HOXB5 biomarker is positively correlated with distant metastases, higher AJCC stage, and poorer prognosis in patients with CRC. HOXB5 expression was an independent and significant risk factor for recurrence and survival in CRC patients. In this same study, the administration of AMD3100, an inhibitor of the upregulation of HOXB5, effectively suppressed HOXB5-mediated CRC metastasis [25].

As shown above, there is an accumulating body of evidence that describes the complex process of development of CRCLM. The primary tumour seems to be involved in creating a favourable microenvironment for tumour seeding. The ECM is an active structure that not only offers a scaffold for metastasis progression, but is also involved in preceding steps of dissemination, such as CTCs survival in the blood stream or seeding in the liver. Treatments directed to interrupt these processes could be effective in the prevention of metastasis. However, further clinical studies will be needed to develop these promising experimental results into effective therapeutic strategies.

## 3. Impact of the Metastasis Molecular Pattern in the Choice of Chemotherapy

There are three possible scenarios in which chemotherapy may play a role in the control of CRCLM: initially unresectable metastatic disease (i.e., metastasis that may become resectable after chemotherapy), resectable metastatic disease and unresectable metastases. For each of these scenarios, the choice of treatment may have a very important impact on the oncological results. For years, fluorouracil was the only active agent for advanced CRC. In the 2000s, the addition of oxaliplatin and irinotecan improved chemotherapy results. Currently, the use of monoclonal antibodies (MCA) has been associated with even better survival results. We will detail the current indications for MCA, related to the molecular profile of the tumour.

### 3.1. Current Indications for MCA

#### 3.1.1. RAS Wild-Type Cancer

In patients with CRCLM, the RAS mutation status permits the selection of patients who are more likely to benefit from treatment with anti-epidermal growth factor receptor (anti-EGFR) agents, such as cetuximab or panitumumab. Approximately 55% of the patients with CRCLM harbour a RAS mutation, the most frequent occurring in exon 2 of K-RAS (42.6%). Mutation at other sites combined (K-RAS exon 3 and 4, and N-RAS exons 2, 3 and 4) have a much lower prevalence [26]. K-RAS is a small G-protein, with a very important role in the EGFR signalling cascade. Mutations in the exon 2 of K-RAS isolate the EFGR pathway and thus render EGFR inhibitors ineffective [27]. There is good concordance of approximately 94% between RAS mutation status in the primary tumour and in distant metastases [28].

Current guidelines recommend extended testing of RAS status, including not only K-RAS exon 2, but also K-RAS exon 3 and 4, and N-RAS exon 2 to 4 [5]. A recent meta-analysis showed that 20% of K-RAS exon 2 wild-type tumours present a mutation in one of the other genes [29]. After analysing eight studies, this meta-analysis concluded that the anti-EGFR therapy was associated with better overall survival (OS) and progression-free survival when all RAS genes were wild-type as compared to the group where only K-RAS exon 2 was wild-type. 

#### 3.1.2. Anti-VEGF Agents

Bevacizumab, a humanized monoclonal antibody that targets vascular endothelial growth factor (VEGF), is the only biologic option for RAS/BRAF mutant tumours. The benefit of adding bevacizumab to standard chemotherapy regimens has been proven, even though, in the context of non-operable metastatic colorectal cancer, the effect on OS and progression-free survival (PFS) is modest. In a pooled analysis from seven RCTs, there was a reduction of 19% in the risk of death for the bevacizumab groups, but with only two months of OS and PFS benefit [30]. Bevacizumab is associated with adverse effects such as bleeding, thrombotic events and bowel perforation, which further places a question mark over its use, giving this modest impact on survival.

#### 3.1.3. BRAF V600 Mutation

This type of mutation is mutually exclusive with RAS mutation and is found in approximately 5–12% of metastatic CRC [31]. According to the meta-analysis of Pietrantonio et al., in patients with RAS WT/BRAF mutation, the addition of anti-EFGR agents does not improve PFS, OS and overall response rate when compared to standard therapy [32]. The study of Cohen et al. also concludes that the anti-EFGR agents are not effective against BRAF V600-mutated CRC metastases [33]. For this reason, current guidelines suggest that a response to either panitumumab of cetuximab of BRAF V600-mutated metastases is unlikely, unless they are given with a BRAF inhibitor [4,5].

#### 3.1.4. Different BRAF Mutations

Non-V600 BRAF mutations occur in 2.2% of CRCLM and are associated with significantly better prognoses than V600 mutations and wild-type BRAF (OS of 60.7 months vs. 11.4 and 43, respectively) [34]. Several studies with a small number of non-V600 BRAF mutations have reported conflicting results related to the use of anti-EFGR MCA [35,36,37]. Therefore, no definitive conclusion related to oncological treatments that target this particular mutation can be drawn.

#### 3.1.5. Mismatch Repair

For patients with unresectable CRCLM and deficient mismatch repair (dMMR), the treatment of choice should be immune checkpoint inhibitor therapy rather than cytotoxic chemotherapy. In the KEYNOTE-177 trial, the use of pembrolizumab in monotherapy when compared with standard chemotherapy was associated with better PFS, response rate and OS [38,39].

#### 3.1.6. HER-2 Amplification

This is seen in approximately 3–4% of patients with CRCLM, especially in RAS-RAF wild-type tumours. It is associated with shorter survival [40,41]. In these types of patients, dual HER2-targeting is recommended, with a combination of trastuzumab with pertuzumab, tucatinib or lapatinib [40].

### 3.2. Initially Unresectable Metastatic Disease

Several randomised control trials have studied, as a primary objective, the impact of the association of MCA with standard chemotherapy regimens in initially unresectable liver metastases (see Table 1).

A meta-analysis from 2012 studied the impact of anti-EFGR agents on patients with initially unresectable K-RAS wild-type CRCLM without any other extrahepatic disease [42]. The study included four RCTs, three of them related to the addition of cetuximab to standard chemotherapy, and the fourth with panitumumab [43,44,45,46]. The authors concluded that the addition of an anti-EFGR agent was associated with a better overall response rate, and improved R0 resection rate and PFS. However, it did not have any impact on OS. The RCT of Ye et al. compared FOLFIRI or FOLFOX standard chemotherapy with and without cetuximab in 138 patients with initially unresectable CRCLM [16]. The study showed a benefit in OS, R0 resection rate and PFS.

The BECOME trial studied the impact of the addition of bevacizumab to FOLFOX in K-RAS mutated patients with colorectal metastases limited to the liver [47]. After analysing 241 patients, there was a statistical improvement using bevacizumab in R0 resection rate and PFS, without benefit in OS.

Two other RCTs also evaluated the impact of MCA, in the context of metastatic disease, but without limiting the results to just CRCLM. The VOLFI study, which examined the impact of the addition of panitumumab to FOLFOX, showed a better response rate and R0 resection index, without showing an improvement in the survival rate [48]. The study of Salz et al. showed that the addition of bevacizumab was associated with better PFS [49].

In conclusion, the majority of RCTs, which focused on the addition of MCA to standard regimens in patients with initially unresectable CRCLM, showed a good response rate and acceptable conversion rates and PFS. However, in all of them, with the exception of the study of Ye et al., statistical differences in the OS of these patients were not achieved.

### 3.3. Resectable Metastases

There is no formal consensus as to whether neoadjuvant chemotherapy is beneficial for patients with initially resectable disease. The type of chemotherapy that should be given in this context is also a matter of debate. However, neither the US nor the European guidelines recommend the use of MCA in the context of perioperative chemotherapy [4,5]. The New EPOC trial, comparing 272 K-RAS wild-type upfront resectable metastases treated with FOLFOX with or without cetuximab, showed that the addition of cetuximab was associated with significantly worse PFS [50,51].

### 3.4. Unresectable Metastases

In the case of unresectable metastases, excluding less frequent mutations (see above), the use of a biological agent may provide some benefit in PFS. The choice of treatment between anti-EGFR and anti-VEGF is based on several aspects, such as RAS/BRAF status (see above), primary tumour site (left vs. right) and the suitability of bevacizumab treatment [52].

In patients with RAS/BRAF mutant tumours, bevacizumab and its analogues offer the only viable choice of MCA. For a RAS/BRAF wild-type tumour, anti-EGFR or anti-VEGF can be chosen depending on the laterality of the tumour and the tolerance of the patient to bevacizumab (see above for adverse effects).

Modest benefits have been reported with the combination of FOLFOX and bevacizumab in non-operable patients [47,49,53,54]. However, as described in the NO 16,966 trial, treatment toxicity is usually associated with its failure more often than with disease progression [49].

Regarding the use of FOLFIRI in combination with bevacizumab, two European trials showed no improvement in survival when compared with FOLFIRI alone [55,56].

For patients with RAS/BRAF wild-type tumours, the use of an anti-EGFR agent with FOLFOX or FOLFIRI is an accepted policy in left-side primary tumours with CRCLM [52]. In the CRYSTAL trial, the use of cetuximab with FOLFIRI was associated with better response rates, PFS and OS [44]. Two single-arm studies have confirmed the efficacy of the combination of panitumumab with FOLFIRI [57,58]. However, there have been no RCTs related to this matter until recently. With regard to the combination of anti-EGFR MCA with oxaliplatin-based regimens, the OPUS and TAILOR trials both showed improvement in the response rate and PFS [43,59]. However, the MRC Coin trial and the NORDIC-VII studies failed to prove a PFS improvement in K-RAS wild-type tumours treated with this combination [45,60].

We summarise the current indications for the administration of treatment related to the molecular characteristics of CRCLM in Figure 1.

## 4. The Impact of the Molecular Profile on Follow-Up

### 4.1. Surveillance for Metastatic Liver Disease

A framework for follow-up is of paramount importance in the management of CRCLM. The early detection of recurrence facilitates early treatment and improves the overall OS. Expert panel recommendations include various clinical parameters, such as testing for Carcinoembryonic Antigen (CEA) levels and CT scans during the first five years after liver resection [4,61]. 

CEA is the only circulating biomarker recommended by the current guidelines for CRC [4,61,62]. Several biomarkers detected in surgical specimens seem to be good prognosis factors. However, their role in the follow-up after resection of primary colorectal cancer or CRCLM has not yet been established. New evidence suggests that combined biomarker clusters will probably be associated with the more accurate prediction of recurrence [63,64].

### 4.2. Circulating Biomarkers

Currently, tissue biopsy continues to be the gold standard for tumour diagnosis. Monitoring tumour progression through repeated tissue biopsies is of limited utility for several reasons—not least because it exposes the patient to repeated invasive procedures. In this context, the concept of the liquid biopsy as a minimally invasive method is promising, not just as a diagnostic tool, but also as a modality for monitoring tumour progression or response to treatment [63,64].

Liquid biopsy is a new technology to detect tumour-related molecular markers in different specimens (e.g., blood, saliva, ascites, stool, urine or cerebrospinal fluid). The most important elements that can be identified in a liquid biopsy are CTCs, circulating free DNA (cfDNA) or circulating tumour DNA (ctDNA), exosomes, tumour-educated platelets (TEP), circulating tumour-derived endothelial cells (CTECs) and protein molecules. Liquid biopsy could become a good strategy for colorectal cancer early detection, monitoring of tumour progression, diagnosis of recurrence or treatment response [63].

In patients with CRC, circulating biomarkers offer attractive ways of diagnosis and monitoring of the disease. To date, there are no specific biomarkers for the follow-up of patients with disseminated CRC. It is worth considering the role of biomarkers in the follow-up of CRC, irrespective of the stage. These biomarkers are categorized into proteins, nucleic acids and CTCs, which differ in availability and cost. Except for CEA, none of the other markers can be currently used in standard clinical practice [63,64].

(a)Proteins

*CEA* is the most widely used and investigated human tumour biomarker. A strong correlation between high serum levels of CEA and cancer progression or metastases recurrence has been documented for CRC [62]. CEA measurement is recommended at 3–6 month intervals for 5 years after liver surgery [4,61]. CEA overexpression of at least 30% is considered significant and should be investigated with an imaging test to detect possible recurrence and for prognosis after surgical resection. CEA elevation alone, without any other relevant test, is inadequate [62,64]. 

*Carbohydrate Antigen 19-9 (CA 19-9)* has been linked with pancreatic and biliary tumours, and to CRC. Recent data showed that CA 19-9, CEA, and RAS or BRAF mutations were associated with a decreased OS. CA 19-9 levels were higher in CRC expressing the BRAF mutation, and this can be useful in the management of these patients [65]. Elevated CA 19-9 levels were associated with worse OS, even though there are no studies that demonstrate the use of CA 19-9 for the follow-up of patients with resected CRCLM [64,66].

*Cancer Antigen 72-4 (CA 72-4)* has been found in different adenocarcinomas such as CRC, gastric, breast and ovarian cancer. The study of Singh et al. showed that the combination of CA 72-4 with TK1 and CEA significantly increases diagnostic sensitivity and specificity for CRC [67]. However, the use of CA 72-4 is not currently justified in the surveillance of resected CRLM [64].

*Survivin* is a member of the inhibitor of apoptosis family and one of the most promising cancer biomarkers. It has the ability to inhibit apoptosis and induce cellular proliferation that ultimately leads to cancer growth and metastasis. A recent review [68] focuses on the development of biosensing systems and biosensors for detecting the survivin protein and its mRNA. In the publication of Ratajczak et al., the authors proposed a new model of interactions of molecular beacons with complementary oligonucleotides, based on the hairpin−hairpin interactions of oligonucleotides, which allows survivin mRNA detection in CCR cells [69]. The same group also showed that a graphene oxide nanosheet can be used as a nanocarrier of molecular beacons for survivin mRNA of CCR [70].

(b)Nucleic Acids

*cfDNA and ctDNA*: cfDNA are derived from normal or tumoural cells undergoing necrosis or apoptosis. Higher levels can be found in cancer patients, because this process is more intense in tumoural tissue. The fraction of cfDNA originating from a tumour is called ctDNA and should contain the same genetic defects as the original tumour cells. The detectable ctDNA can be used to detect mutations such as RAS or EGFR. Plasma and urine cfDNA levels are higher in metastatic disease than in healthy patients, which can be used to monitor disease development [63].

Kobayashi et al. showed that patients with resectable CRCLM who had preoperative detectable ctDNA in their plasma had statistically lower DFS and a tendency to have lower OS [71]. Achieving undetectable levels of ctDNA after CRCLM resection seems also to be associated with better DFS and a good pathological response to neoadjuvant chemotherapy [63,72,73]. Tie at al observed a 40-fold decrease in mutated ctDNA levels after neoadjuvant chemotherapy. However, this decrease was not associated with improved DFS [73].

At present, it seems that both pre- and postoperative levels of ctDNA are associated with the histological response to treatment and with the post-resection prognosis of CRCLM patients. It seems likely that liquid biopsy with ctDNA detection will become an increasingly useful tool for CRCLM diagnosis and follow-up.

(c)Exosomes and microvesicles

Recent studies have shown that intercellular communication can be mediated through so-called extracellular vesicles (EVs). EVs are cellular products that serve as intercellular vehicles for membrane and cytosolic proteins, lipids and RNA. The presence of these EVs can also be used as a tumour marker. EVs may be divided in two categories: exosomes (derived from the multivesicular endosome, e.g., from the cell interior) and microvesicles (derived from the cellular membrane) [74].

*Exosomes* have the capacity to transport a variety of substances, including proteins related to membrane transport and various nucleic acids, such as microRNAs (miRNAs), long non-coding RNA (IncRNA), circular RNA (circRNA), transfer RNA (tRNA), small nuclear RNA (snRNA) and small nucleolar RNA (snoRNA). At present, there are some cancer immunotherapy studies which are based on exosome detection and measurement for diagnostic, prognostic, treatment sensibility and predictive purposes in CRC [63,75].

*miRNAs* are small sequences of nucleotides which are involved in the regulation of gene expression, including oncogenes or tumour suppressors, but which can also regulate the tumour microenvironment [63]. miRNAs dysregulation has been associated with various cancers such as CRC [63,64].

A particularly interesting characteristic of miRNAs is their capacity to differentiate between metastatic and non-metastatic CRC. The study of Hu et al. demonstrated that mir-1229 levels were statistically higher in TNM (tumour node metastasis cancer classification) stages III and IV and that its levels corelated with poor survival. Furthermore, the same study showed that the same miRNA played a role in activating the VEGF pathway and that inhibiting its production also inhibited angiogenesis [76]. In another experimental study, the levels of miR-17-5p and miR-92a-3p were also higher in metastatic disease [77]. The studies of Yan et al. and Peng et al. encountered decreased levels in serum exosomes of miR-638 and miR-548c-5p in patients with CRCLM when compared with patients with localized CRC disease [78,79]. Therefore, we might predict that in the near future, a battery of several miRNAs might be highly sensitive and specific for patients with CRCLM and that this type of test could serve as a non-invasive biomarker in post-resection follow-up.

Exosomal miRNA, IncRNA, CircRNA can be considered promising biomarkers. Despite the fact that several studies have confirmed a potential role in the development and evolution of CRC, further studies are needed in order to establish their clinical applicability [63,64].

(d)Circulating cells

*Tumour-educated platelets (TEP).* Platelets are the second most abundant peripheral blood cell with a well-established role in haemostasis and thrombosis. Cancer cells can activate platelets by direct interaction or through several released mediators [80]. Direct interaction of platelets with tumour cells seems to be essential for cancer progression. The study of Petersen et al. demonstrates that VEGF, PDGF (platelet-derived growth factor) and platelet Factor 4 were elevated in platelets of CRC patients [81].

To date, the contributions of TEP to CRC progression are still not clear, and their role as a tumoural marker needs more study. The study of Qian et al. found that the pre- and post-treatment ratio of mean platelet volume in non-metastatic CRC was a prognostic factor for OS [82]. Even more interestingly, the study of Yang et al. identified TIMP mRNA in platelets as a potential independent diagnostic biomarker for colorectal cancer. The same study showed that the platelets can carry this mRNA, a stimulus for cancer development, inside tumoural cells [83].

*CTCs* are cells that are detectable from the peripheral blood samples of cancer patients and most metastatic patients. CTC detection in the early stages of CRC is challenging, and its utility in this scenario remains limited [63]. However, CTCs have been identified as a potential non-invasive diagnostic and prognostic marker for CRLM. In a meta-analysis of 12 studies from 2013, OS and PFS were demonstrated to be worse in CRLM patients who were positive for CTCs [84]. CTCs may also improve diagnostic accuracy, with sensitivity of 83%, and of 91.5%, when combined with CEA level [85]. In a study of patients after CRLM resection, no correlation was found between the presence of CTCs in peripheral blood and early recurrence [86]. However, at least two studies showed that CTC detection is more sensitive in portal circulation than in peripheral blood. Furthermore, both studies showed a correlation between CTC levels in portal circulation with prognosis in resected CRCLM patients, which was not present in peripheral blood [87,88]. Nonetheless, CTCs are not yet recommended in routine clinical practice and have not been investigated in patients with resected CRLM as a follow-up marker [63,64].

## 5. Conclusions

Understanding the molecular behaviour of CRCLM is of vital importance. Many important steps have already been made in the selection of chemotherapy according to the molecular pattern of the tumours, especially in the case of RAS and BRAF mutations. The main impact of these treatments in their current form is in transforming initially unresectable metastases into resectable disease. However, the effect on OS in these patients is still modest.

Recent studies have shown the complexity of the process of CRCLM seeding. We believe that in the coming years, translational research should focus on this subject. Many mechanisms related to the process of metastatic seeding could be modified through molecular treatments, preventing the early steps in the dissemination of cancer.

One of the great challenges in the follow-up of patients with resected CRCLM is the identification of patients with higher probabilities of early recurrence and the early diagnosis of this recurrence. As described above, several molecular tools available in liquid biopsy are showing promising results. However, we believe that more clinical studies are needed before implementing these resources in the protocols of CRCLM follow-up.

## Figures and Tables

**Figure 1 ijms-24-01127-f001:**
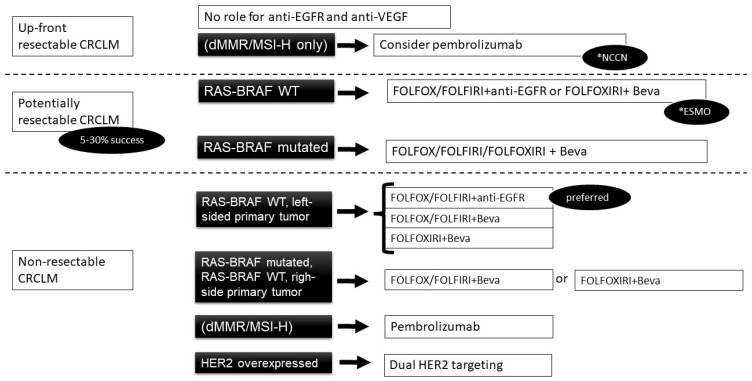
Algorithm of the role of molecular characteristics in the choice of treatment in CRCLM (*—as described in the specified guidelines).

**Table 1 ijms-24-01127-t001:** Impact of ACM on initially unresectable diseases.

**With Cetuximab**												
Name	DOI	Treatment	KRAS WT LLD patients	Follow-up	PFS	*p*	OS	*p*	Resection R0	*p*	Treat response	*p*
OPUS 2011	10.1093/annonc/mdq632	FOLFOX-Cetuxi vs. FOLFOX	48(25/23)		HR 0.64	0.39	HR 0.93	0.85	16% vs. 4.3% (RR 3.68)	0.23	RR 1.94	0.02
CRYSTAL 2011, Van Cutsem	10.1200/JCO.2010.33.5091	FOLFIRI-Cetuxi vs. FOLFIRI	140 (68/72)	29.7 Mo	HR 0.56	0.04	HR 0.85	0.43	13.2% vs. 5.5% (RR 2.38)	0.13	RR 1.59	0.003
MRC Coin 2011	10.1016/S0140-6736(11)60613-2	CAPOX or FOLFOX-Cetuxi vs. CAPOX or FOLFOX	178 (87/91)	21 Mo	HR 0.68	0.03	NR	NR	15% vs. 13% (RR 1.13)	0.74	NR	NR
Le-Chi Ye 2013	10.1200/JCO.2012.44.8308	FOLFIRI or FOLFOX- Cetuxi vs. FOLFIRI of FOLFOX	70/68	25 Mo	HR 0.60	0.004	HR 0.54	0.013	25.7% vs. 7.4% (OR 4.37)	0.004	57.1% vs. 29.4%	0.001
**With Panitumumab**												
Name	DOI	Treatment	KRAS WT LLD patients	Follow-up	PFS	*p*	OS	*p*	Resection R0	*p*	Treat response	*p*
Douillard 2010	10.1200/JCO.2009.27.4860	FOLFOX-Pani vs. FOLFOX	116 (60/56)	13.2 Mo	HR 0.82	0.43	HR 0.93	0.81	27.8% vs. 17.5% (RR 1.59)	0.19	NR	NR
**With Bevacizumab**												
Name	DOI	Treatment	KRAS mutated LLD patients	Follow-up	PFS	*p*	OS	*p*	Resection R0	*p*	Treat response	*p*
BECOME	10.1200/JCO.20.174	FOLFOX+Beva vs. FOLFOX	241 (121/120)	37 Mo	HR 0.49	0.001	HR 0.71	0.31	22.3% vs. 5.8%	0.01	54.5% vs. 36.7%	0.001
**Conversion Chemotherapy without Limited Liver Disease**												
Name	DOI	Treatment	KRAS WT patients	Follow-up	PFS	*p*	OS	*p*	Resection	*p*	Treat response	*p*
VOLFI	10.1200/JCO.1901340	FOLFOXIRI-Pani vs. FOLFOXIRI	96 (63/33)	44.2 Mo vs. 63.3 Mo	HR 1.07	0.76	HR 0.67	0.12	33% vs. 12.1% (OR 3.63)	0.02	OR 4.469	0.004
Salz 2008	10.1200/JCO.2007.14.9930	CAPOX or FOLOFOX-Beva vs. CAPOX or FOLFOX	700 vs. 701 (no KRAS specified)	15.6 Mo	HR 0.83	0.0023	HR 0.89	0.769	8.4% vs. 6.1%	NR	OR 0.9	0.31

## Data Availability

No new data were created or analysed in this study. Data sharing is not applicable to this article.

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
