# Peer review of "The Impact of Molecular Biology in the Seeding, Treatment Choices and Follow-Up of Colorectal Cancer Liver Metastases—A Narrative Review"

_ijms, 2023, doi:10.3390/ijms24021127_

Round 1
Reviewer 1 Report
In this paper, the Authors have reviewed the complex molecular pathways involved in the colorectal cancer liver metastases (CRCLM). The review is focused on the dissemination of malignant cells, the molecular pattern associated with the chemotherapy selection as well as the response to cancer treatment and predictive biomarkers. First, the role of the extracellular matrix (ECM) in the spreading of cancer was indicated. The Authors have focused on the circulating tumor cells (CTCs), their capacity for seeding in a distant organ and changes produced in the cellular and extracellular composition of the organ. The Authors have also assessed the recent progress in the application of monoclonal antibodies (MCA) in anti-cancer therapies. The effects of MCA addition to standard chemotherapy regiments on overall survival (OS), progression-free survival (PFS) and overall response rate are shown. The Authors also discuss the challenges of the application of liquid biopsy as the monitoring tool for cancer diagnostics, tumor progression, and response to treatment. The roles of circulating free DNA (cfDNA), circulating tumor DNA (ctDNA), exosomes, tumor-educated platelet (TEP), circulating tumor-derived endothelial cells (CTECs), and protein molecules are emphasized.
This review will be invaluable for doctors and scientists involved in cancer therapy and in the development of safe and effective cancer treatments. I recommend the paper for publication after minor revision addressing the issues listed below.
1. Figure 1 caption should be rephrased.
2. The words “tumor”/”tumour” should be in one consistent form in the entire text.
3. A discussion of the role of protein Survivin and its mRNA in cancer progression and metastasis is missing. These important markers have a significant impact on the disease prognosis and patient survival (see recent review: Biosensors and Bioelectronics 2019, 137, 58-71, and papers: ACS Appl. Mater. Interfaces 2018, 10, 20, 17028–17039; Nanomaterials 2018, 8(7), 510). These relevant literature references should be cited.
4. It would be beneficial for the General Readership to define and explain the abbreviations used in the manuscript (e.g., TNF, PGDF, snRNA).
5. Line 215: “at al” should be “et al”.
Reviewer 2 Report
Thank you for a very well written review. One question:
1. You say that no European or American guidelines propose neoadjuvant chemotherapy for upfront resectable CRCLMs. However the ESMO guidelines state that although one can go directly to surgery, based on the RCT from Nordlinger et al. Lancet 2013, neoadjuvant treatment improves DFS. Do you want to comment on that?
Author Response
Please, see the attachement
